# Key elements of a successful integrated community-based approach aimed at reducing socioeconomic health inequalities in the Netherlands: A qualitative study

Lisa Wilderink[1,2]*, Ingrid Bakker[2], Albertine J. Schuit[3], Jacob C. Seidell[1], Carry M. Renders[1,2]

1 Department of Health Sciences, Faculty of Sciences, Amsterdam Public Health research institute, Vrije Universiteit Amsterdam, Amsterdam, The Netherlands, 2 Department of Healthy Society, Windesheim University of Applied Sciences, Zwolle, The Netherlands, 3 School of Social and Behavioral Sciences, Tilburg University, Tilburg, The Netherlands

* l.wilderink@vu.nl

**Data Availability Statement:** All qualitative data files are available from the Data Archiving and Networking Services (DANS) repository database

## Abstract

### Background

Since 2010, the Zwolle Healthy City approach, an integrated community-based approach, has been implemented in the Dutch municipality of Zwolle. This approach is proven successful in reducing health inequalities. However, the key elements of this approach are not clear. The current study aimed to identify key elements of this successful local community-based approach, according to the perspectives of various stakeholders.

### Methods

Semi-structured interviews were carried out with 29 professionals who were involved in the approach in the period 2010–2018 and have occupations at the strategic ($n = 4$), tactical ($n = 17$) and operational level ($n = 8$). Data was analyzed using the thematic analysis approach.

### Results

We identified nine perceived key elements that contributed to the success of the approach aimed at reducing socioeconomic health inequalities. The respondents indicated the following key elements: (1) collaboration between a variety of local organizations that want to have impact on the health of citizens; (2) support for the approach on the strategic, tactical and operational level of involved organizations; (3) proper communication and coordination, both for the network and within the organizations; (4) embeddedness in organizations' policies and processes and (5) collaboration with private organizations is of added value, although there is no "one size fits all". Other key elements are (6) collaboration with citizens, (7) profiling the approach like a brand and (8) moving along with and taking advantage of opportunities. Finally, (9) continuous monitoring and evaluating goals and processes, and learning from the results, is important.

and can be found via https://easy.dans.knaw.nl/ui/datasets/id/easy-dataset:181161 https://doi.org/10.17026/dans-zkf-46xc.

**Funding:** This study was supported by ZonMw, Den Haag, the Netherlands (project number 531001314). The funders had no role in study design, data collection and analysis, decision to publish, or preparation of the manuscript. https://www.zonmw.nl/nl/.

**Competing interests:** The authors have declared that no competing interests exist.

## Conclusion

Nine key elements were identified that, according to various stakeholders, contributed to the success of the Zwolle Healthy City approach. These insights are important to further strengthen the Zwolle Healthy City approach but may also help and inspire other local integrated community-based approaches aimed at reducing socioeconomic health inequalities, to improve and adapt the approach within their specific local and dynamic context.

## Background

Unhealthy dietary and physical activity behaviors are more prevalent in populations with a low socioeconomic position than in populations with a high socioeconomic position [1]. These unhealthy behavioral patterns contribute to a lower perceived health, higher prevalence of overweight and higher incidence of non-communicable diseases. Hence, people with a low socioeconomic position on average have a lower (healthy) life expectancy [2–4].

These socioeconomic health inequalities are caused by numerous factors on the individual level that influence healthy behavior negatively, e.g. psychological distress, lower self-efficacy for behavioral change, a lack of knowledge about healthy choices and a lower level of health literacy [5–8]. There is an increasing awareness, however, that, in addition to those individual factors, socioeconomic health inequalities are largely due to a wide range of unfavorable circumstances in the social and physical environment. The unfavorable environment in underprivileged neighborhoods offers less support and fewer opportunities for healthy behavior [9–12]. Populations with a low socioeconomic position often live in neighborhoods with, for example, less public park space to be physically active, more marketing of unhealthy products and a higher density of fast-food restaurants [13,14]. Social characteristics of the neighborhood, such as social safety and social cohesion, also have a negative influence on outdoor physical activity behavior [15].

Due to this complex combination of factors contributing to socioeconomic health inequalities, policy and practice increasingly recognize that reducing health inequalities requires integrated action across different intervention levels (individual, community and society) [16,17]. Upstream community-based and multilevel approaches aimed at addressing factors in the social and physical environment are the most likely to reduce socioeconomic health inequalities [18].

Socioeconomic health inequalities are a challenge for health policy and society in the Netherlands as well [19,20]. Dutch municipalities are encouraged by the national government to implement the so-called JOGG approach, which aims to encourage healthy dietary and physical activity behavior and prevent overweight in children in a healthy environment. JOGG (the Dutch acronym for Youth at a Healthy Weight) is an integrated community-based approach, based on the French EPODE approach [21]. This complex community-based approach targets multiple sectors and multiple levels of influence: interpersonal, organizational, community and policy. The JOGG approach includes a variety of interventions that target individual, social and physical environmental determinants of overweight and obesity. Characteristic of this community-based approach is that it is implemented by multiple organizations in the community, both within and outside the health domain, coordinated locally by a program manager [22].

Since 2010, several local policy and community-based organizations in the Dutch municipality of Zwolle have been committed to the approach in order to stimulate healthy behavior,

which made Zwolle the first Dutch JOGG municipality. This local approach, called 'Zwolle Healthy City' has proven successful as regards stimulating healthy behavior and reducing socioeconomic health inequalities: the prevalence of overweight in children in the two neighborhoods with the lowest socioeconomic position decreased more in comparison to other neighborhoods in Zwolle [23]. Today, more than one-third of all municipalities in the Netherlands have implemented the JOGG approach.

Zwolle Healthy City and also JOGG approaches implemented in other municipalities have been evaluated and in some cases show success, as they show encouraging changes in stimulating healthy behavior. In several JOGG municipalities even lower percentages of overweight were found or socioeconomic health inequalities were reduced as in Zwolle [23,24]. However, the key elements of these approaches, and why they contribute to success, are often unclear. Moreover, in previous studies little attention was paid to the long-term development of elements since evaluation mostly takes place after a relatively short period of time.

The aim of the current study is, therefore, to identify key elements of the Zwolle Healthy City approach–a successful integrated community-based approach aimed at reducing socioeconomic health inequalities–according to the perspectives of various professionals. We do so by reflecting on the long-term implementation of the approach (eight years), which enhances understanding of the key elements of the approach so that they can be improved. Additionally, by identifying these key elements, other comparable programs aiming at reducing socioeconomic health inequalities or similar can take inspiration from them in their specific local context.

## Methods

### Design

To identify the perceived key elements of the successful Zwolle Healthy City approach, a qualitative study was performed. The study protocol was approved by the Medical Ethics Review Committee (METc) of VU University Medical Center, registered with the US Office for Human Research Protections (OHRP) as IRB00002991. The METc, chaired by prof. dr. C. Boer, confirmed the study did not require medical ethical approval under Dutch legislation on medical trials (2018.601). All procedures performed were in accordance with the WMA declaration of Helsinki.

Semi-structured interviews were carried out with professionals from different organizations and backgrounds into their experiences with the implementation and execution of the approach during the period they were involved (during the entire period of 2010–2018, or part thereof).

The organizations involved are the local government, the regional public health service, a university of applied sciences, two welfare organizations, a home care organization and a municipal sports service. After analyzing the data of the semi-structured interviews, a focus group discussion with a selection of the respondents was held to verify the conclusions. See S1 File for more information on how COREQ requirements were met.

### Setting

The central approach in this study is 'Zwolle Healthy City', an integrated community-based approach aimed at reducing socioeconomic health inequalities, which has been implemented since 2010 in the medium-sized municipality of Zwolle (around 125,000 citizens) in a rural area in the northeast of the Netherlands. In Zwolle, 41 percent of the households form the lowest income group, which is comparable to the country's average (40 percent) [25]. 80 percent of the adults inhabitants in Zwolle experience their health as '(very) good', 18 percent

experience it as 'okay' and 2 percent as '(very) bad', which is also comparable to the national averages. 46 percent of the inhabitants that experience their health as '(very) bad' have trouble making the financial ends meet.

In 2005, the Dutch government stimulated that cities would focus on preventing and counteracting overweight in children aged 0–19 years (in the context of the 'Large cities Policy'). In the municipality of Zwolle, this led to a community-based approach in two disadvantaged neighborhoods. The percentage of households that form the lowest income group are in those *neighborhoods* 60 and 65 percent. From 2006 through 2009, many health-promoting activities were implemented in both neighborhoods, without structural local coordination.

In 2010, the municipality of Zwolle decided to tackle the prevention of overweight in children in a more integrated and systematic way, inspired by the EPODE approach developed in France [21]. In the Netherlands, this approach has been called the JOGG approach since then. The approach, focused on improving the lifestyles 'healthy nutrition' and 'healthy PA', was implemented in the municipality of Zwolle, building on the existing community-based approach in the disadvantaged neighborhoods, with some minor adjustments appropriate to the local context. This Zwolle Healthy City approach focused on preventing overweight in children aged 0–19 years living in two disadvantaged neighborhoods.

Since 2014, the Zwolle Healthy City approach has extended its focus, content-wise as well as with respect to the target population, by also including senior citizens [26]. The two disadvantaged neighborhoods were extended with three additional areas. Today, the Zwolle Healthy City approach stimulates healthy behavior by directing towards a healthy lifestyle, healthy social & physical environment and healthy care in disadvantaged neighborhoods, thereby aiming to reduce socioeconomic health inequalities (see Fig 1 for the logic model). Examples of what the approach constitutes content wise can be found in S3 File.

## Study population

Respondents were purposively sampled to create a heterogeneous interview sample (*n* = 29, 12 male, 17 female). Purposive sampling is a goal-oriented sampling method in which the respondents are chosen based on the judgment of the researchers (LW, IB) [27]. Respondents were

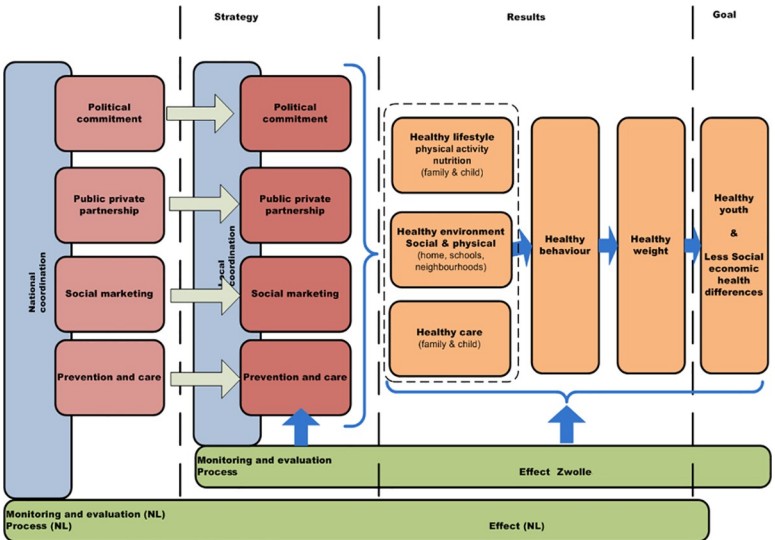

**Fig 1. Logic model of the Zwolle Healthy City approach.**

selected based on the organization they work for, their occupation level and their contribution to the implementation of the approach in the years 2010–2018.

To recruit respondents at the tactical level of involved organizations, all members of the program group were invited to participate. The program group consists of professionals working for the organizations involved as manager, policy adviser or (associate) professor. They meet monthly to share and discuss the implementation of the approach. All twenty (previous) members of the program group since 2010 were invited to participate in the study; seventeen of whom agreed to participate in an interview. Reasons for non-participation were (a) the feeling of not being able to contribute, (b) not feeling able to remember much and (c) personal circumstances. To collect a wider perspective on the implementation, the program group members were asked which professionals were involved in the approach at the operational level as sports club advisor, community sports worker, child worker, youth worker or community worker. Based on this, eleven community workers were asked to participate in an interview; eight of whom agreed to participate. Reasons for non-participation were (a) emigration and (b) the feeling of not being able to contribute. Additionally, six municipal executive councilors who had been involved in the approach since the start in 2010 and work at the strategic level, were asked to participate in an interview; five of whom agreed to participate. The reason for non-participation is unknown. All respondents were invited via email (by LW) and, in case of no response, contacted by telephone.

Table 1 lists the organizations the respondents work for and how many of them have occupations at the strategic ($n = 4$), tactical ($n = 17$) and operational level ($n = 8$). The average number of years that respondents were involved in the approach is 5.6 years in the period 2010–2018.

## Data collection

Respondents were interviewed (by LW) in the period from January to March 2019. In a one-to-one interview, they were asked about their experiences with the Zwolle Healthy City approach. The duration of the interviews ranged from 49 to 92 minutes. The last four interviews did not reveal new information, which meant data-saturation was reached. Written informed consent was obtained from the respondents beforehand.

The interview guide was drawn up on the basis of the logic model of the Zwolle Healthy City approach (Fig 1), literature on community-based approaches and documented conversations with involved professionals in 2013 (partly the same professionals). The interview guide can be found in S2 File. The guide consisted of open-ended and follow-up questions structured into the following topics: local structure and coordination, support, interventions and activities, aligning prevention and healthcare, monitoring and evaluation, collaboration, and citizen participation. The interview guide was discussed within the project research team (LW, IB, AJS, JCS, CR).

**Table 1. Overview of involved organizations and the number of respondents that work for that organization.** n = 29.

|  | Strategic | Tactical | Operational | Total |
|---|---|---|---|---|
| Local government | 4 | 2 | 0 | 6 |
| University of applied sciences | 0 | 4 | 0 | 4 |
| Regional public health service | 0 | 2 | 0 | 2 |
| Municipal sports service | 0 | 2 | 3 | 5 |
| Welfare organization | 0 | 4 | 5 | 9 |
| Home care organization | 0 | 3 | 0 | 3 |
| **Total** | 4 | 17 | 8 | **29** |

After analyzing the data of the semi-structured interviews, 7 of the 29 interviewed professionals participated in a verifying focus group discussion. Again, these respondents were purposively sampled to select a representative group for the complete sample in terms of organizations and occupation levels. Four of them work at the tactical level of involved organizations, three at the operational. Respondents were asked to reflect on the report of the key elements that were extracted from analysis of the semi-structured interviews. The respondents agreed with the preliminary conclusions and only had a few suggestions for minor adjustments on the text were made afterwards. These adjustments were made to align the conclusions with the experiences of the respondents. The adjustments did not change the content of the conclusions, but only structure and language.

## Data analysis

The interviews were audio recorded and transcribed verbatim anonymously. MAXQDA 2018 was used to analyze the data. Data was analyzed in a systematic way, using the thematic analysis approach with the aim to identify returning themes [28]. To ensure the reliability of data interpretations, interviews were open coded separately by two researchers (LW, DW), by labelling fragments with descriptive codes. Differences were discussed and agreed upon. After open coding five interviews, axial coding took place by creating codes that reflect multiple fragments and codes were grouped together in themes into a coding scheme. The project research team (LW, IB, AJS, JCS, CR) discussed the coding scheme. The coding scheme functioned as a base for the analysis of the following interviews. The coding scheme and themes were regularly discussed by two researchers (LW, DW). Selective coding took place by constantly comparing codes and finding correlations between codes. The derived themes formed the base of the key elements.

## Results

The 29 professionals who participated in this qualitative study mentioned a wide range of factors that contributed to the success of the Zwolle Healthy City approach. Based on the semi-structured interviews and focus group discussion, nine key elements of the approach were identified. The key elements and their supporting sub-themes are described in Table 2 and explained in the following sections, with illustrative quotes.

### Collaboration between a variety of local organizations

According to the respondents, it is important to collaborate with a varied, cross-domain group of organizations that, despite their differences in e.g. mission, tasks and methods, have a common goal. This variety in organizations is typical for an integrated community-based approach that focuses on creating a supportive social and physical environment that enhances healthy behavior from multiple levels and settings. Organizations that can and want to do something for the health and well-being of people in the neighborhood should be involved, e.g. care and welfare organizations, municipal health and sports services and the local government.

> *"It is actually a "coalition of the willing", of people who want to join and are able to contribute a lot. (. . .) the organizations themselves must understand the importance. And as long as you see that, you participate and you commit yourself. You would actually like to see a very broad coalition of parties that matter".* Professional at the tactical level #7.

According to the respondents, it is helpful that these organizations operate as an independent network that decides its own agenda, in addition to executing 'basic tasks' from the local

**Table 2. Overview of key elements with supporting sub-themes.**

| Key element | Supporting subthemes |
|---|---|
| *Collaboration between a variety of local organizations* | |
| | Shared goals despite differences in e.g. mission, tasks and methods |
| | Coalition of the willing |
| | Aligning practices and using each other's expertise |
| | Collaboration at the operational level as well |
| *Support on three levels: strategic, tactical, operational* | |
| | Broad political support |
| | Mutual gains approach |
| | Communicating with professionals at operational level |
| *Communication and coordination* | |
| | Leader for network (program manager) |
| | Vertical alignment within organizations |
| | Leader within each organization (internal coordinator) |
| *Embeddedness of the approach in organization's policy and processes* | |
| | Embedded in policies and regular tasks and working processes |
| | Transition from program to movement |
| *Collaboration with private organizations* | |
| | Financial resources, knowledge and decisiveness |
| | Ask private organizations to pursue a specific goal |
| | Leaving role open |
| *Collaboration with citizens* | |
| | Involve citizens in small steps and not too long |
| | Citizen involvement from the start |
| | Convert citizens' input quickly into action |
| *Profiling the approach like a brand* | |
| | Raising awareness among organizations |
| | Linking success stories to approach |
| | Less important for citizens |
| *Move along with, and take advantage of, (local and national) opportunities* | |
| | Building on with "what is already there" |
| | Aligning approach with regular working processes |
| | Link up with national opportunities |
| *Continuous monitoring and evaluating goals and processes, and learning from the results* | |
| | Measuring outcome effects |
| | Monitoring process |
| | University of applied sciences involved |

government that most of the involved organizations have. This gives the program group a sense of autonomy. At the same time, two remarks can be made here. Firstly, some respondents indicate that there is a risk of false equivalence in the network because of the dual role that the local government has: as an equal network partner and as the commissioning party. This can lead to confusion about roles and independence, which does not benefit collaboration. Secondly, one respondent indicates that due to the non-hierarchical structure, a sense of a clear task is sometimes lacking for the members of the network, which can be paralyzing. This means that some guidance is desirable.

Furthermore, in addition to collaboration at the tactical level in a network to align practices (the program group), it is important that multiple organizations also collaborate at the

operational level. According to the respondents, professionals from different organizations should run projects together and use each other's expertise.

## Support on three levels: Strategic, tactical and operational

Respondents describe three levels of all involved organizations that need support for the integrated community-based approach. Firstly, the Zwolle Healthy City approach should be widely supported at the strategic level within the local governmental organization (broad political support) and at the strategic level of other involved organizations. In the past eight years of implementation, municipal executive councilors from the public health domain and other domains (e.g. sports, urban planning and the social domain) were involved in the approach and thereby work towards a healthy city.

> *"This can never be the sole responsibility of the public health executive councilor. If we do it, we must support it with all the executive councilors, because this is also about infrastructure. Can children cycle to school safely? It's about playing in parks, so it's about the public space. It is about safety on the street, can you be outside safely, walking, exercising. It's actually about all the portfolios".* Professional at the strategic level #1.

Secondly, to gain support at the tactical level from the organizations involved, it is important to act according to the mutual gains approach: not only should the organization's involvement be meaningful for the approach, but there should also be a 'win' for the organization itself.

Thirdly, support for the Zwolle Healthy City approach at the operational level is important among the professionals who run the projects and are in contact with the citizens on a daily basis. One respondent said it is important to have good communication between the professionals working at the tactical and operational level about the goals of the approach and what those goals mean to the professionals in their daily work. Knowing why tasks must be performed and what the relevance is for their work, contributes positively to the support and motivation among professionals to achieve goals that are set.

> *"In that program group, they say all kinds of things, but we actually don't know what those things imply for our work".* Professional at the operational level #3.

In addition to these professionals' need to be involved in decisions made at a higher level, it is important that they experience support to encourage healthy behavior. This support must be more than just the commitment in available working hours. It also involves being given the opportunity to develop necessary skills and to experience moral support from supervisors and other colleagues. Encouraging healthy behavior among citizens must be considered important and must be supported and facilitated by the organization, according to the respondents.

## Communication and coordination

Respondents indicate someone should be appointed to coordinate the network. In the Zwolle Healthy City approach, this is the task of the program manager who, inter alia, organizes meetings for the program group and reminds the organizations involved of their responsibilities with regard to the goals of the approach. According to the respondents, it is, therefore, important that the role of the program manager is independent of the organization he or she works for. The respondents see the fact that the program manager of the Zwolle Healthy City approach is not employed by the local government as a success factor of the approach.

*"I think* [that it's a good thing that the program manager does not work for the local government] *because a certain tension can arise about the local government, which is a partner across the whole, but also the commissioning party".* Professional at the tactical level #7.

Besides support for the approach at the strategic, tactical and operational level, there must be vertical alignment: the respondents indicated that alignment and communication between those three levels within each organization are important. In addition, coordination is not only important to the network as a whole, but also within the involved organizations. An appointed internal coordinator within the organization can take a leading role when it comes to integrating the theme 'healthy behavior' into the daily work.

*"You should really have someone who is responsible for that theme* [healthy lifestyle], *because otherwise it will just become a side issue. (. . .) It is the task of that person to inform the rest of the colleagues and to get them onboard on that theme".* Professional at the operational level #2.

This internal coordinator serves multiple purposes, depending on the organization he or she works for. The common purpose of the internal coordinator is to bridge the gap between the tactical level of the approach (the program group) and the professional who work at the operational level. He or she supports professionals in paying attention to healthy lifestyle, providing concrete information on how to do this and inform colleagues about goals and ambitions of the approach.

## Embeddedness of the approach in organization's policy and processes

According to the respondents, in the first three years of the implementation, the approach was seen as a program with financed projects. Subsequently, it is now seen just as a *'movement, multiple organizations with a common goal'* (Professional at the tactical level #3). In addition to the network of organizations for connecting expertise, the approach is mainly embedded in the policies and regular processes of the organizations involved. In other words, health is considered in every activity or event, for example by offering healthy food and drinks.

*"Before, it was obviously a program that also set the agenda. It was about getting organizations to focus on promoting health. Now you see that the focus is there, but we're trying to find the connection and how to use each other's expertise".* Professional at the tactical level #6.

However, the risk of embedding the approach in regular processes of the organizations is that attention for stimulating healthy behavior drops, because of more priority for other themes. For that reason, despite embedding, it is important to designate a coordinator in each of the organizations who is still responsible for ongoing attention among colleagues for the healthy behavior theme, according to the respondents.

*"It should be someone who becomes the driving force behind it. We quickly fall back to the daily routines and before you know it, you're like,* [colleague's name] *was going to do something with it, but it did not happen".* Professional at the operational level #3.

Furthermore, respondents indicate that since the approach has been embedded among the involved organization, the local government no longer provides funding for activities but only for their core business, the 'basic tasks'. Consequently, professionals experience being limited to spending fewer hours, resulting in a reduced impact on the mission of the Zwolle Healthy

City approach. Professionals feel that with these additional tasks for the approach, they have to do more within the current agreements and funding.

## Collaboration with private organizations

All respondents agree that collaboration with private (local) organizations can be of added value to the approach. One respondent describes it as *'a bonus, not a basis'*. The bonus could be financial resources, knowledge, and more decisiveness, according to the respondents. Private organizations have a different working method than public organizations, i.e. they think more often in terms of action. At the same time, differences in culture, language and interests can be difficult when collaborating.

> *"They* [private organizations] *are really like, well if we want something, then. . . chop-chop, action. So that is a different pace"*. Professional at the tactical level #1.

Within the respondent group, two different views exist on collaborating with private organizations. Some respondents state that it is important to ask private organizations with a certain expertise to pursue a specific goal in a project. Other respondents, on the other hand, believe that private organizations put up more resistance when their role is already fully specified, and it's therefore better to let private organizations decide for themselves how they can contribute to the goals of the approach.

> *You just have to say*: *"We are Zwolle Healthy City and this is our mission and these are our goals. Would you like to help us in this*? *And then they really come up with very brilliant ideas sometimes"*. Professional at the tactical level #8.

## Collaboration with citizens

Many respondents mention the importance of collaboration with citizens in aligning the approach with citizens' wishes and needs. So far, the Zwolle Healthy City approach has only done this in the organization and execution of specific interventions. This participation varies from drinking coffee with citizens to hear their wishes to starting a trajectory in which citizens come up with ideas which they implement and evaluate themselves. When organizing an event or running a project, the respondents consider it important to let citizens think along from the start and if possible throughout the whole process, i.e. in the design, organization, execution and evaluation. For example, with the Club2Move intervention, youngsters can exercise twice a week and participate in activities around healthy nutrition in a community sports club. The youngsters participate in promotion, organization and execution of the activities. Another example is when children and parents were invited to design, construct and maintain a natural playground in the neighborhood. It's important to convert their input into action quickly, in the professionals' experiences. Moreover, to make citizen participation as successful as possible, you have to involve citizens in small steps and, moreover, processes should not take too long.

> *"You should not expect that you will get them along in a long development and organization process, that will also make you very unhappy"*. Professional at the tactical level #14.

One should not completely outsource the organization of activities to the citizens; a facilitating role of a professional is necessary, since citizens often want to contribute, but not fully organize and be responsible.

*"So, we managed to let people help with the organization, but it was not possible to put the organization completely into the hands of those citizens. And they didn't want to take that responsibility either"*. Professional at the operational level #8.

## Profiling the approach like a brand

The respondents indicated that bringing the approach to people's attention is especially important for raising awareness and commitment in the municipality and a sense of devotedness among the network organizations. One respondent said that an informative website about the approach contributes positively to profiling the approach like a brand, which is not the case in the municipality of Zwolle anymore. In the first three years of implementation, resources were available for external communication, which led to available hours for a communication advisor and a communication plan. By profiling the approach like a brand, success stories can be linked to the approach, which encourages other organizations to also want to be part of it.

*"Often- For example, the municipal sports service organizes something great for young people or senior citizens, and it is wonderful and successful, but it is communicated outwards as something of the municipal sports service, and there is hardly any link made to Zwolle Healthy City. That makes me think that the approach is actually successful, but that it is not clear to the outside world"*. Professional at the tactical level #4.

On the other hand, some of the respondents indicate that it does not matter to the citizens whether an activity is part of the approach or not.

*"You don't have to profile the approach like a brand for the citizens, in my experience. I think it is more like a quality label for the network, for the collaborations, for the organizations. Like, "Do you want to be a part of it*?*" Professional at the tactical level #2.

## Move along with, and take advantage of, (local and national) opportunities

According to the respondents, the aim to align with "what is already there" is typical of the approach. The Zwolle Healthy City approach achieved this in three ways. Firstly, the approach was built up from a (smaller) network of organizations on a rather similar mission (but different scope), that was already there. As a result, some of the professionals involved already knew each other from previous collaborations. This made the start in 2010 slightly easier. Secondly, the approach tries to align with regular tasks, activities and working processes as much as possible. Thirdly, the local approach tries to link up with national developments. A national focus on a certain topic (e.g. the importance of drinking water or healthy school communities), can provide resources and impetus for the local implementation.

*"The healthy school approach is something that has been stimulated nationwide for 6/7 years now, and we link up with that to ensure that schools are healthy and that it's not a random project that will stop. I think it is nice that we have the wind in our sails with the fact that we can align with existing subsidy programs"*. Professional at the tactical level #2.

However, according to the respondents, you always have to consider what you need locally, because national programs do not always fit the local situation.

*"It is also important that you can say no to things, and that you say*: *"Well, fantastic that there is so much money, but that does not work at all for that neighborhood–the issues that we*

*see, so. And then, before you know, you have lost a lot of energy on something that does not even fit, that you have to justify, that gives you hassle"*. Professional at the tactical level #5.

## Continuous monitoring and evaluating goals and processes, and learning from the results

All respondents state that monitoring and evaluation of goals is important. They say it is important to show development in the objectives to keep involved organizations committed, and show them what they are contributing to. However, according to one respondent, the outcome effects that you measure (e.g. the percentage of children that is overweight) cannot simply be clarified by the activities and interventions undertaken *"because there are all sorts of other factors that play a role"* (Professional at the tactical level #3).

In addition to monitoring and evaluating the goals, monitoring and evaluating the process are just as important, according to the respondents. This allows the process to be continuously adjusted, newly derived topics to be put on the agenda and discussions about "where are we now and how do we move on" to provide insight and, therefore, a certain 'grip' on the approach. It is important to use these evaluations to improve activities and processes. If the evaluation shows that it is not working properly, the network of organizations should agree on doing things differently.

The fact that one of the partners of the Zwolle Healthy City approach is a research group from a university of applied sciences reinforces this key element. The researchers are responsible for monitoring the approach and students with various backgrounds are also involved. They do exploratory research on questions concerning a 'healthy city', raised by involved professionals or institutions, which stimulates efficient implementation.

## Discussion

Nine perceived key elements were identified that contributed to the success of Zwolle Healthy City, an integrated community-based approach aimed at reducing socioeconomic health inequalities. Additionally, we identified factors reinforcing or hindering this success.

Literature on key elements of integrated approaches aimed at reducing socioeconomic health inequalities is limited. Nevertheless, Van Koperen et al. [29] describe, in line with our results, the importance of a broad and cross-domain group of organizations that all contribute to a stated mission, and the British National Institute for Health and Clinical Excellence (NICE) also describes the need for a multi-agency approach in their guidance for the prevention of overweight [30]. Our results further support the idea stated by the Dutch National Institute for Public Health and the Environment (RIVM), that in addition to professionals working at the strategic and tactical level, professionals from different organizations working at the operational level should also collaborate [31]. In contrast to our results, NICE does not explicitly describe the need for political support at the strategic level [30]. RIVM on the other hand, states that for a successful approach, the local government should be in charge of directing collaboration between various policy sectors and cooperation partners [31]. The key elements about collaborating with private organizations and citizens correspond to earlier overviews in the literature [29,31,32]. In a study into successful determinants of public-private partnerships, Leenaars et al. [33] stress the added value of collaborating with private organizations for financial contribution, manpower and knowledge. Our results about the importance of collaborating with citizens in order to align the approach with citizens' wishes, needs, talents and motivations is in line with recent research on community engagement [34]. O'Mara-Eves et al. [35] even conclude in a meta-analysis that community engagement has a positive impact on health outcomes for disadvantaged groups.

In 2010, the Zwolle Healthy City approach started with a set of expected critical components derived from the EPODE approach in France that led to components in the logic model, functioning as guidance for the approach. The respondents in this study indicate some of those components as key elements. Interestingly, this is not the case for 'integrating prevention and care'. The approach invested in strengthening the connection between prevention and care on a project basis, and could be seen as a promising key element. Nonetheless, the respondents do not see this as a successful key element, since due to national policy changes in 2015 the attention for strengthening the connection from the perspective of the local government dropped. However, according to the literature, individual targeted prevention is a proven expedient for overweight citizens [36–38].

In addition, key elements that cannot be found in the current logic model either are about collaboration between a variety of local organizations, support for the approach on the strategic, tactical and operational level (which is broader than 'political commitment'), collaboration with citizens throughout the entire process, profiling the approach like a brand, moving along with and taking advantage of opportunities, and embeddedness of the approach in organization's policy and processes. Moreover, the logic model now shows a 'strategy' column; however, the identified key elements are relevant at the tactical and operational level as well.

## Strengths and limitations

Interviewing a varied group of professionals about eight years of implementation made it possible to identify perceived key elements for successful implementation of a community-based approach aimed at reducing socioeconomic health inequalities. Nonetheless, limitations of this study should be mentioned. Firstly, the design was retrospective, which means that the respondents had to retrieve experiences from their memory. This may have led to loss of details and muddled memory with the passage of time, since the first phase of implementation was more than eight years ago. Moreover, because the approach has been proven successful, respondents might mainly look back on the positive sides of the implementation and play down negative sides. To decrease this drawback, available documented conversations with (a selection of) the same professionals in 2013 were read by the interviewer (LW) beforehand, who could, therefore, help respondents to remember details of implementation in the first three years.

One strength of this study is that a considerable part of the professionals who contributed to the approach in 2010–2018 were invited for an interview and agreed to participate, and that they were employed by various organizations and on different occupation levels. This made it possible to formulate key elements from the perspective of the strategic, tactical and operational level. Including 29 respondents was sufficient because data saturation was reached after approximately 25 interviews. Another strength is that the respondents have been involved in the approach for a relatively long period of time (see Table 1), allowing them to give a good impression.

## Implications for practice and research

The overview of key elements described in this study may help and inspire other local integrated community-based approaches aimed at socioeconomic health inequalities or similar. Nowadays, more than one third of all Dutch municipality's implement the JOGG approach, which the Zwolle Healthy City approach is largely based on. Professionals working in those municipalities can use the results in this study to improve the implementation process of this comprehensive approach. These results show which aspects on the process level are worth investing in, the key elements, and what factors contribute to strengthening those aspects. This applies not only to Dutch initiatives but also to other local integrated approaches worldwide.

Involved professionals working on the local Zwolle Healthy City approach can learn from the results as well, since the key elements provide guidance for improvement of the Zwolle Healthy City approach and accessory logic model. Further strengthening of the approach can take place based on the conclusions, and in collaboration with citizens, because our results indicate that citizens should be involved from the start and throughout the entire process, i.e. in the design, implementation and evaluation of activities.

The Zwolle Healthy City approach targets multiple sectors (e.g. public health, welfare, spatial planning, traffic and transport) and multiple levels of influence (interpersonal, organizational, community and policy). This type of local multi-level approach has long been the standard for tackling complex issues such as obesity and socioeconomic health inequalities. Recent literature has increasingly been paying attention to the relationships between those different levels in a whole-system approach [39–41]. In a whole-system approach, the central concern is to focus on the relationships between individuals and organizations working within a system [32]. The complexity of this type of approach lies not only in the relationships between the different levels, but also in how those relationships work in a certain context. Attwood et al. [42] state that we should not look at approaches as "one size fits all" but consider the issues within their wider environmental context. Questions as 'how does it work' and 'in what context does it work' are just as important as 'what works'. For the evaluation of an approach in complex system thinking, this implicates that it is not only important to describe what the key elements of an approach are, but also which dynamic processes the impact of the key elements can reinforce or hinder (i.e. positive and negative feedback loops operating in the system).

For this reason, the main suggestion for further research is to investigate the mechanisms associated with the key elements and describe the context in which these mechanisms work.

## Conclusions

This study provides an overview of nine key elements that, according to stakeholders, contributed to the success of the Zwolle Healthy City approach, an integrated community-based approach aimed at reducing socioeconomic health inequalities. The key elements contain collaboration between a variety of local organizations, support for the approach on the strategic, tactical and operational level, communication and coordination, collaboration with private organizations, collaboration with citizens, profiling the approach like a brand, moving along with and taking advantage of opportunities, embeddedness of the approach in organization's policy and processes, and continuous monitoring and evaluating goals and processes, and learning from the results.

The results enable further development of the Zwolle Healthy City approach, by improving the implementation of the key elements as they provide guidance for policy making and activities. Apart from further strengthening the Zwolle Healthy City approach these insights may also help and inspire other integrated community-based approaches aimed at socioeconomic health inequalities or similar, to improve and adapt the key elements within their specific local and dynamic context. More research on mechanisms associated with the key elements and the context in which these mechanisms work is needed.

## Supporting information

**S1 File. Consolidated criteria for reporting qualitative studies (COREQ) 32-item checklist.**
(DOCX)

**S2 File. Interview guide–key elements of a successful local integrated community-based approach.**
(DOCX)

**S3 File. Examples from practice.**
(DOCX)

## Acknowledgments

We would like to thank all respondents who participated in this study. The authors would like to thank Diana Wilmink for helping to analyze the data.

## Author Contributions

**Conceptualization:** Lisa Wilderink, Ingrid Bakker, Albertine J. Schuit, Jacob C. Seidell, Carry M. Renders.

**Data curation:** Lisa Wilderink.

**Formal analysis:** Lisa Wilderink.

**Funding acquisition:** Ingrid Bakker, Albertine J. Schuit, Jacob C. Seidell, Carry M. Renders.

**Investigation:** Lisa Wilderink.

**Methodology:** Lisa Wilderink, Ingrid Bakker, Albertine J. Schuit, Jacob C. Seidell, Carry M. Renders.

**Project administration:** Ingrid Bakker.

**Supervision:** Ingrid Bakker, Albertine J. Schuit, Jacob C. Seidell, Carry M. Renders.

**Writing – original draft:** Lisa Wilderink.

**Writing – review & editing:** Lisa Wilderink, Ingrid Bakker, Albertine J. Schuit, Jacob C. Seidell, Carry M. Renders.

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
