## [Decision Letter · Decision Letter 0]

22 Jul 2020

PONE-D-20-14827

Key elements of a successful integrated community-based approach aimed at reducing socioeconomic health inequalities in the Netherlands: a qualitative study

PLOS ONE

Dear Dr. Wilderink,

Thank you for submitting your manuscript to PLOS ONE. After careful consideration, we feel that it has merit but does not fully meet PLOS ONE’s publication criteria as it currently stands. Therefore, we invite you to submit a revised version of the manuscript that addresses the points raised during the review process.

We look forward to receiving your revised manuscript.

Kind regards,

William Joe

Academic Editor

PLOS ONE

Journal Requirements:

Reviewers' comments:

Reviewer's Responses to Questions

**Comments to the Author**

1. Is the manuscript technically sound, and do the data support the conclusions?

Reviewer #1: Yes

Reviewer #2: Partly

2. Has the statistical analysis been performed appropriately and rigorously? 

Reviewer #1: Yes

Reviewer #2: N/A

3. Have the authors made all data underlying the findings in their manuscript fully available?

Reviewer #1: Yes

Reviewer #2: No

4. Is the manuscript presented in an intelligible fashion and written in standard English?

Reviewer #1: Yes

Reviewer #2: Yes

5. Review Comments to the Author

Reviewer #1: This is a very well written paper. It is technically sound and presented in a very readable form. The font style needs to be consistent throughout the paper. The references are in a different font. Else, the paper seemed acceptable to me. Thank you

Reviewer #2: Multi-sectoral interventions are increasingly viewed as critical to reducing health inequities improving preventive interventions, and reducing the burden of noncommunicable diseases. However, examples of successful implementation of such intervention is limited. This manuscript seeks to fill this gap. The use of qualitative data is very helpful for process evaluations of implementation projects, and this paper uses qualitative data to describe the factors that contributed to the success of the Zwolle Healthy City approach. Although this paper contributes information on a very important topic, the significance of the findings and applicability to other similar projects was not completely clear. In addition, the rationale and presentation of qualitative findings could use revision. Please see below for specific areas that could be improved.

- More details on the Healthy City program, including a description of what the specific interventions are that constitute the integrated community-based approach would be helpful.

- Additional information on the local context and a brief description of Zwolle City would also be helpful.

- Minor comment: In the introduction, there are too many acronyms. It would be better to just spell out SEP and SEHI.

- I think it is unnecessary (and could potentially allow participants to be identified) to list each one with their gender, role, organization, and years of participation. A summary table would be more appropriate.

- The logic behind the research design and order of procedures is a little bit unclear and could use some clarification. Specifically, it would be helpful to know why and how the first round of interviews in 2013 were conducted, and something about those participants, unless that data has already been published?

- The presentation of the qualitative results would be more effective if quotes were placed in-text with the themes and sub-themes they are associated with. Table 2 would be more effective with the themes, sub-themes, and brief definitions of these.

- The treatment of some of the themes/sub-themes is a little superficial. Specifically, additional details on the specific coordination mechanisms employed and how specifically citizens collaborated with the intervention/partnership would improve the results.

- The alphanumeric codes assigned to individuals are not very meaningful by themselves. Rather, it would be more helpful to have some descriptor of their role (i.e., community worker 1, etc.).

- The 2013 interviews and later FGD data do not appear in the results. They should either be added (or if already there, then it should be clearer what they contribute), or dropped from the methods.

- Although it is not necessary for qualitative data to be broadly generalizable in the same way that quantitative data can be, some further discussion of the contribution of this work, and generalizability of the Zwolle City experience to other parts of the Netherlands, or other sites would improve the manuscript. As is, the significance of the findings seems a little limited in scope.

6. PLOS authors have the option to publish the peer review history of their article (what does this mean?). If published, this will include your full peer review and any attached files.

Reviewer #1: **Yes: **Poulami Dasgupta

Reviewer #2: No

---

## [Author Response · Author response to Decision Letter 0]

7 Sep 2020

August 21, 2020

Dear editor and reviewers, 

Thank you for taking the time to read our paper and providing it with feedback. In the table below our response to the comments are shown. We hope to respond sufficiently to the feedback points raised.

Page numbers mentioned refer to pages in the revised manuscript with track changes.

Sincerely,

Lisa Wilderink, MSc

Style is adjusted, PLOS ONE style requirements are now met.

The track changes concerning the style requirements are not visible in the file, to improve readability.

2. We note that you have indicated that data from this study are available upon request. PLOS only allows data to be available upon request if there are legal or ethical restrictions on sharing data publicly. For information on unacceptable data access restrictions, please see 

The minimal anonymized data set is uploaded to the Data Archiving and Networking Services (DANS) repository and can be found via https://easy.dans.knaw.nl/ui/datasets/id/easy-dataset:181161

https://doi.org/10.17026/dans-zkf-46xc

3. Please include captions for your Supporting Information files at the end of your manuscript, and update any in-text citations to match accordingly. Please see our Supporting Information guidelines for more information: 

Captions for Supporting Information are now listed at the end of the manuscript and in-text citations match accordingly.

4. This is a very well written paper. It is technically sound and presented in a very readable form. The font style needs to be consistent throughout the paper. The references are in a different font. Else, the paper seemed acceptable to me. Thank you

Thank you. References have been adjusted and are in the same font as the manuscript now.

5. Multi-sectoral interventions are increasingly viewed as critical to reducing health inequities improving preventive interventions, and reducing the burden of noncommunicable diseases. However, examples of successful implementation of such intervention is limited. This manuscript seeks to fill this gap. The use of qualitative data is very helpful for process evaluations of implementation projects, and this paper uses qualitative data to describe the factors that contributed to the success of the Zwolle Healthy City approach. Although this paper contributes information on a very important topic, the significance of the findings and applicability to other similar projects was not completely clear. In addition, the rationale and presentation of qualitative findings could use revision. Please see below for specific areas that could be improved. 

See below for the areas that are improved.

6. More details on the Healthy City program, including a description of what the specific interventions are that constitute the integrated community-based approach would be helpful.

We understand that it is helpful to have insight in the content of the approach. For that reason, Supporting Information file 3 shows examples of interventions and projects that were carried out. To make this more clear for the reader, we now refer to the Supporting Information file 3 (p.7).

7. Additional information on the local context and a brief description of Zwolle City would also be helpful.

We have added information on health and income of the inhabitants of Zwolle to provide some more insight in the local context (p.6). Moreover we have added supporting information about the specific interventions that constitute the integrated community-based approach (see supporting file 3)

8. Minor comment: In the introduction, there are too many acronyms. It would be better to just spell out SEP and SEHI.

SEP and SEHI are now spelled out in the manuscript.

9. I think it is unnecessary (and could potentially allow participants to be identified) to list each one with their gender, role, organization, and years of participation. A summary table would be more appropriate.

The original table is removed and a summary table with an overview of the organizations is added (p.8). The roles of respondents, gender and the average number of years involved are now described in the text (p.7-8).

10. The logic behind the research design and order of procedures is a little bit unclear and could use some clarification. Specifically, it would be helpful to know why and how the first round of interviews in 2013 were conducted, and something about those participants, unless that data has already been published?

We understand that introducing the 2013 interviews causes unclarity. For that reason, we do not mention those interviews anymore in the methods section. The 2013 interviews did not lead to the results in this current study.

11. The presentation of the qualitative results would be more effective if quotes were placed in-text with the themes and sub-themes they are associated with. Table 2 would be more effective with the themes, sub-themes, and brief definitions of these.

Quotes are now placed in-text with the themes and follow in the paragraphs after mentioning the subtheme. 

12. The treatment of some of the themes/sub-themes is a little superficial. Specifically, additional details on the specific coordination mechanisms employed and how specifically citizens collaborated with the intervention/partnership would improve the results.

Additional details on coordination (p.17) and collaboration with citizens (p.19) is added in the text.

13. The alphanumeric codes assigned to individuals are not very meaningful by themselves. Rather, it would be more helpful to have some descriptor of their role (i.e., community worker 1, etc.).

Describing the specific role could jeopardize the anonymity. However, to meet your recommendation we now refer to the respondents as ‘professional working at the strategic/tactical/operational level’. 

14. The 2013 interviews and later FGD data do not appear in the results. They should either be added (or if already there, then it should be clearer what they contribute), or dropped from the methods.

The 2013 interviews are not included in the results. For that reason, they are dropped from the methods.

The function of the FGD was to reflect on the results that were extracted from analysis of the semi-structured interviews. As mentioned on page 10, the respondents agreed with the preliminary conclusions and only a few minor adjustments were made afterwards. Adjustments were made to align the conclusions with the experiences of the respondents and did not concern the content of the conclusions, but only structure and language.

15. Although it is not necessary for qualitative data to be broadly generalizable in the same way that quantitative data can be, some further discussion of the contribution of this work, and generalizability of the Zwolle City experience to other parts of the Netherlands, or other sites would improve the manuscript. As is, the significance of the findings seems a little limited in scope.

In the revised manuscript we elaborate more on the significance of the findings for Dutch municipalities which implement the quite similar JOGG approach (145 in total), and comparable approaches worldwide (p.25).

---

## [Editor Report · Decision Letter 1]

2 Oct 2020

Key elements of a successful integrated community-based approach aimed at reducing socioeconomic health inequalities in the Netherlands: a qualitative study

PONE-D-20-14827R1

Dear Dr. Wilderink,

We’re pleased to inform you that your manuscript has been judged scientifically suitable for publication and will be formally accepted for publication once it meets all outstanding technical requirements.

Kind regards,

William Joe

Academic Editor

PLOS ONE
---

## [Editor Report · Acceptance letter]

8 Oct 2020

PONE-D-20-14827R1 

Key elements of a successful integrated community-based approach aimed at reducing socioeconomic health inequalities in the Netherlands: a qualitative study 

Dear Dr. Wilderink:

I'm pleased to inform you that your manuscript has been deemed suitable for publication in PLOS ONE. Congratulations! Your manuscript is now with our production department. 

Kind regards, 

on behalf of

Dr. William Joe 

Academic Editor

PLOS ONE